# p27^kip1^ Modulates the Morphology and Phagocytic Activity of Microglia

**DOI:** 10.3390/ijms231810432

**Published:** 2022-09-09

**Authors:** Jolien Beeken, Sofie Kessels, Jean-Michel Rigo, Yeranddy A. Alpizar, Laurent Nguyen, Bert Brône

**Affiliations:** 1UHasselt, Hasselt University, BIOMED, 3500 Hasselt, Belgium; 2Laboratory of Molecular Regulation of Neurogenesis, GIGA-Stem Cells, Interdisciplinary Cluster for Applied Genoproteomics (GIGA-R), University of Liège, C.H.U. Sar-Tilman, 4000 Liège, Belgium

**Keywords:** p27^kip1^, microglia, cell migration, process motility, phagocytosis, morphology

## Abstract

p27^kip1^ is a multifunctional protein that promotes cell cycle exit by blocking the activity of cyclin/cyclin-dependent kinase complexes as well as migration and motility via signaling pathways that converge on the actin and microtubule cytoskeleton. Despite the broad characterization of p27^kip1^ function in neural cells, little is known about its relevance in microglia. Here, we studied the role of p27^kip1^ in microglia using a combination of in vitro and in situ approaches. While the loss of p27^kip1^ did not affect microglial density in the cerebral cortex, it altered their morphological complexity in situ. However, despite the presence of p27^kip1^ in microglial processes, as shown by immunofluorescence in cultured cells, loss of p27^kip1^ did not change microglial process motility and extension after applying laser-induced brain damage in cortical brain slices. Primary microglia lacking p27^kip1^ showed increased phagocytic uptake of synaptosomes, while a cell cycle dead variant negatively affected phagocytosis. These findings indicate that p27^kip1^ plays specific roles in microglia.

## 1. Introduction

Microglia, the immunocompetent cells of the brain, are key players in brain development and homeostasis [1,2,3]. These cells have multiple tasks, including the clearance of dying neurons, the modulation of neurogenesis and neuronal activity, the promotion of angiogenesis, and the control of blood–brain barrier integrity [4,5]. To fulfill this extensive range of tasks, microglia appear highly dynamic with a complex morphology. This morphology becomes gradually more intricate during differentiation in the developing brain [6]. In a healthy brain, microglia are ramified with a small cell body and long, thin processes that continuously survey the surrounding brain parenchyma [3,7,8,9]. This patrolling activity is important to monitor, sense, and regulate neuronal activity during homeostasis but also to rapidly detect potentially harmful insults [10,11]. When pathologically triggered, microglia transit into a reactive state, characterized by increased proliferation and the production of reactive oxygen species and pro-inflammatory cytokines [12,13,14]. They rapidly extend processes with enlarged and bulbous tips towards the damaged region and revert to their ‘ground’ state when brain homeostasis is restored [2,15,16]. Microglial morphology, branch motility, and migration are dependent on cytoskeletal rearrangements; however, the intracellular mechanisms steering these processes remain unclear [9,17,18].

One candidate protein for regulating microglial cytoskeletal remodeling, and consequently their immunological and homeostatic functions, is p27^kip1^ (hereafter referred to as p27). p27 is a 198 amino acid-long protein containing distinct cyclin and cyclin-dependent kinase (CDK) binding sites at its N-terminus and interaction sites for RhoA and a nuclear localization signal at its C-terminus [19]. A comparison between mouse and human p27 sequences revealed a 92% overlap, including several phosphorylation sites and other functional motifs such as RhoA binding sites. This sequence conservation suggests similar functions for p27 in mice and humans [20]. p27 has been first identified as a cell cycle inhibitor of the Cip/Kip protein family. It binds and inhibits cyclins and cyclin-dependent kinases and is therefore well studied as a tumor suppressor protein in both mice and humans [21,22,23,24,25,26,27,28]. While the cell cycle inhibitory role of p27 has been extensively described in different types of human cancer, other studies question the beneficial role of p27 and show a reduction in nuclear p27 and a gain in cytoplasmic p27 after oncogenic stimuli, thereby increasing tumor progression and malignancy [20,28,29]. In addition to its cell cycle function, p27 controls cell migration and morphology. Cells lacking p27 often show an amoeboid shape with reduced morphological complexity [30,31,32]. In human cancer cells, mouse fibroblasts, and epithelial cells, p27 shuttles from the nucleus to the cytoplasm to modulate migration [23,29]. In mice, cell migration is characterized by cytoskeletal rearrangements, and both the actin and microtubule (MT) cytoskeleton are regulated by p27 [23,30]. p27 controls actomyosin reorganization by inhibiting RhoA activation, whereas it binds to and promotes MT polymerization whether or not by interfering with stathmin activity [19,23,30,33,34]. p27 is also involved in the stabilization and accumulation of Neurogenin-2 in cortical progenitors, a protein that promotes the specification and migration of cortical neurons [34,35,36,37,38]. Furthermore, the p27 protein modulates axonal transport independently from its cell cycle function [39].

Despite the wide range of properties of p27 in other human and mouse cell types, research on microglia only states a decrease in p27 expression after brain-penetrating injury [40]. Yet, little is known about the relevance of the protein for microglial dynamics and functions. Here, we explore the role of p27 in microglia using a combination of in vitro and in situ approaches, with a focus on branch motility, migration, and phagocytosis.

## 2. Results

### 2.1. p27^kip1^ Is Expressed in Microglial Cytoplasm and Regulates Microglial Morphology

Tight regulation of the microglial cytoskeleton is crucial for fulfilling its broad range of immunological and homeostatic functions [17,41]. The MT-associated protein (MAP) p27 controls cytoskeleton-related functions in other cell types [38]. Therefore, we analyzed two different mouse models, namely a p27 knockout (p27KO) and a p27 cell cycle dead variant (p27CK^−^), to test whether p27 also plays a function in microglia. The p27 cell cycle dead variant cannot bind cyclins and cyclin-dependent kinases (CDKs), and yet, other functional domains of the p27 protein, such as the RhoA and Rac binding domains, remain intact [42,43]. First, we confirmed p27 expression in p27WT and p27CK^−^ primary microglia while the absence of p27 was observed in p27KO microglia (Figure 1a). Additionally, RT-PCR shows the expression of p27 in a murine microglial cell line (BV2), in primary microglia (PM), and in a whole brain (WB) extract (Figure 1b). Immunolabeling of primary microglia (Iba1+) revealed the enrichment of p27 in the soma around the nucleus and in the cell processes (Figure 1c). The p27CK^−^ variant is slightly more stable, due to disturbed degradation, and is therefore known to accumulate more than wild type p27 in fibroblasts [42,43]. This accumulation seems to be present in primary microglia as well (Figure 1a,c). The cytoplasmic enrichment of p27 might suggest a putative role in regulating the microglial cytoskeleton and, hereby, the control of microglial morphology. Interestingly, primary p27KO and p27CK^−^ microglia seem less ramified compared to p27WT microglia, which suggests a reactive state (Figure 1c). This altered morphology was, however, not related to changes in the microglial proliferation rate, quantified as the number of Ki-67+ microglia after 24 h in culture (Appendix A). Additionally, the expression of p27 upon microglial pro- and anti-inflammatory stimulation, and the production of nitric oxide upon IFNɣ stimulation was unchanged (Appendix A). This likely suggests that p27 does not regulate the microglial immune response.

As microglia show a broad range of highly complex morphologies in vivo, microglial structural complexity was studied in fixed brain slices of young adolescent (P21) CX3CR1^eGFP/+^ p27WT, CX3CR1^eGFP/+^ p27KO and CX3CR1^eGFP/+^ p27CK^−^ mice to assess the effect of p27 on microglial morphology. Microglial process complexity was analyzed by performing a three-dimensional Sholl analysis [44,45], from which the outcoming cell skeletons of p27KO and p27CK^−^ microglia showed an aberrant reduction in morphological complexity (Figure 2a and Appendix A). The total process length was significantly reduced in p27KO and p27CK^−^ microglia (Figure 2b and Appendix A). Furthermore, p27KO and p27CK^−^ microglia displayed a reduced ramification (Figure 2b and Appendix A), and fewer process endings or tips (Figure 2d and Appendix A). Unlike p27KO microglia (Figure 2e), p27CK^−^ microglia also showed a reduced number of branch points (Appendix A). The observed morphological changes do not result from a disturbed density of microglia in p27KO and p27CK^−^ brains, as microglial numbers are similar between genotypes (Figure 2f,g, Appendix A).

Taken together, our results show that p27 is expressed in microglial cells, and p27KO and p27CK^−^ microglia exhibit reduced morphological complexity. However, microglia of p27-deficient mice show a normal density in the brain. These findings suggest that p27 may modulate the dynamic features of microglia, likely depending on the expression level of p27, thereby revealing a possible role for p27 in microglial branch motility which accounts for their surveilling capacity in the brain.

### 2.2. Microglial Brain Surveillance Is Not Altered upon Loss of p27^kip1^ or Knock-In of a Cell Cycle Dead Variant of p27^kip1^

As p27KO and p27CK^−^ microglia show a less ramified morphology compared to p27WT microglia in situ, we postulated a role for p27 in microglial brain surveillance. We performed time-lapse recordings of eGFP-expressing microglia in acute brain slices obtained from young adolescent (P21) CX3CR1^eGFP/+^ p27WT, CX3CR1^eGFP/+^ p27KO and CX3CR1^eGFP/+^ p27CK^−^ mice using two-photon microscopy and quantified the surveilled area in the cortex. Overlaying time-consecutive images revealed highly dynamic microglia continuously scanning the environment over a time span of 10 min (Figure 3a). Both depletion of p27 (p27KO, Figure 3b), and expression of the cell cycle dead variant (p27CK^−^, Figure 3c) did not affect brain surveillance by microglial cells.

### 2.3. Conditional Loss of p27^kip1^ Alters Specific Parameters of Microglial Migration Ex Vivo

Cell migration and axonal transport are disrupted in neurons of p27KO mice [19,24,38,39]. As microglial migration and branch motility are highly affected by neuronal activity, we cannot exclude that the genetic ablation of p27 in newborn neurons affects microglial migration parameters [11,46]. To assess whether p27 plays a cell-intrinsic role in microglia, we generated a conditional knockout (cKO) mouse where p27 was specifically eliminated in microglia upon subcutaneous injection of tamoxifen in pregnant females (p27^Fl/+^CX3CR1^CreEr(T2)/+).^ Recombination efficiency was assessed by RT-PCR with primers located outside the loxP sites (Appendix A). Successive tamoxifen injection at E12.5 and E13.5 eliminated p27 expression in microglia by E15.5 (Appendix A). To investigate microglial migration within the timeframe of ongoing neuronal migration, we used two-photon confocal microscopy on acute brain slices from E15.5 embryos of p27cKO and p27WT littermates [6,47]. We tracked microglia migrating over a time span of 6 h and quantified migratory parameters (Figure 4a). Only cells remaining in the field of view for at least 100 min were included in the analysis. Tracking analysis revealed no significant differences in the average migration speed of microglia in the cortical parenchyma of p27cKO and p27WT mice (Figure 4b), suggesting that p27 is not required for the intrinsic control of microglial migration. Migration of primary p27WT, p27KO, and p27CK^−^ microglia was also not significantly different in an in vitro scratch migration assay (Appendix A). However, microglia do not migrate at a constant speed. Instead, they continuously alternate between scanning their environment without soma displacement, also known as idling, and displacement of the soma in the direction of one process while retracting the others, resulting in a saltatory migration pattern [3,48,49]. To investigate the role of p27 in this saltatory motion, the instantaneous speed, i.e., the speed between consecutive time points, was plotted as a function of time for three p27WT and p27cKO microglia with a high, intermediate, and low average speed, respectively, as previously described by our lab (Figure 4c) [50]. A threshold for idling was calculated from the mean squared displacement using a custom-made macro developed by Gorelik and Gautreau and set at 1.5 µm per min [49]. The relative idling time, defined as the percentage of time the cell spent on pausing, did not differ between p27WT and p27cKO microglia (Figure 4d). However, the mean instantaneous speed of events below the idling threshold was significantly higher in p27cKO microglia (Figure 4e). On the other hand, the mean instantaneous speed of the active migration events above the idling threshold was significantly lower in p27cKO microglia, as compared to p27WT microglia (Figure 4e). Yet, only cells with a high average active migration speed (>2.5 µm/min) contributed to this difference (Figure 4f).

Taken together, our results show that the average microglial migration speed is not altered upon conditional loss of p27. The time spent idling was not affected, but the active migration speed was significantly lower in p27cKO microglia compared to their p27WT counterparts, suggesting a possible migration delay upon loss of p27.

### 2.4. Process Extension Speed towards a Cortical Lesion Is Decreased upon Expression of the Cell Cycle Dead Variant of p27^kip1^

Homeostatic brain surveillance by microglia was unchanged upon lack of p27 expression or expression of its cell cycle dead variant (p27CK^−^) (Figure 3). However, changes in brain homeostasis (e.g., ATP release by dying cells) can activate a wide array of membrane receptors present on microglial processes that trigger directed process motility towards the damaged region [2,7,15,51,52]. In order to test whether p27 contributes to this process, we performed laser-induced brain injury on CX3CR1^eGFP/+^ p27WT, CX3CR1^eGFP/+^ p27KO, and CX3CR1^eGFP/+^ p27CK^−^ cortical brain slices using two-photon confocal microscopy to mimic brain damage associated with cell death and abundant release of chemical cues such as ATP [7]. We quantified the average and instantaneous speed of process extension per cell over a total duration of 15 min. Upon injury, all surrounding microglia extended their processes to the damaged region within 15 min at a comparable average process extension speed between p27WT and p27KO microglia (Figure 5a,b,d). On the other hand, p27CK^−^ microglia exhibited a slower extension speed as compared to p27WT cells (Figure 5a,c,e).

Altogether, these data suggest that the loss of p27 does not affect microglial directed process motility, while the expression of its cell cycle dead variant, which is known to be more stable due to disturbed degradation, slows down microglial process extension towards a brain lesion.

### 2.5. The Phagocytic Capacity of Microglia Is Altered In Vitro upon Loss of p27^kip1^ or Knock-In of Its Cell Cycle Dead Variant

Directed process motility by microglia towards damaged tissue or weak synapses tagged for elimination is often followed by phagocytosis [16,53,54,55]. To evaluate whether p27 may influence the phagocytic capacity of microglia, we investigated synaptosome uptake in p27WT, p27KO, and p27CK^−^ microglia. We incubated primary microglia with fluorescently labeled synaptosomes and assessed the percentage of phagocytic microglia at different time points (from 10 min to 2 h) by means of cell fluorescence following synaptosome uptake. In addition, we analyzed fluorescence intensity differences to determine the amount of synaptosome uptake in phagocytic microglia. Loss of p27 increased the percentage of phagocytic microglia after 2 h of incubation (Figure 6a), whereas expression of p27CK^−^ decreased the percentage of phagocytic cells (Figure 6c). Yet, the amount of engulfed synaptosomes, quantified by the median fluorescent intensity (MFI) was not significantly altered among the genotypes (Figure 6b,d).

Our results suggest a dosage-dependent role of p27, i.e., an effect depending on the expression level of p27, on the phagocytic capacity of microglia.

## 3. Discussion

Microglia, the resident immune cells of the CNS, play key roles during brain development and homeostasis [3,56]. Malfunction of microglia is correlated with developmental and neurodegenerative disorders, emphasizing the importance of proper microglial function and regulation. Here, we tested whether p27 plays a role in microglial cell biology and function. Our study in mice offers insight into the specific roles of p27 in microglial functions. In particular, depletion or dysfunction of p27 specifically affects microglial morphological complexity as well as their phagocytic activity and directed process motility.

### 3.1. p27^kip1^ May Act as a Regulator of Morphological Complexity without Modulating Microglial Brain Surveillance or Inflammatory Response

As microglia are the immune cells of the brain, we first examined whether p27 regulates microglial inflammatory responses such as proliferation and nitric oxide production. We did not observe altered p27 expression levels upon microglial stimulation, and the constitutive absence of p27 did not induce increased microglial proliferation, nor did it alter nitric oxide production (Appendix A). This is in line with research demonstrating that the depletion of p21^cip1^, another cell cycle protein, does not affect the proliferation of microglia and astrocytes before and after serum mitogenic stimulation [57]. p21^cip1^ is strongly expressed in microglia and is known to drive NO and TNFα release upon microglial activation. Therefore, p21^cip1^ might be responsible for a compensatory effect in p27KO and p27CK^−^ microglia, hence preserving normal nitric oxide production [57]. Taken together, our results suggest that p27 is dispensable for the microglial inflammatory response.

Our results point to a role for p27 in the regulation of the morphology of microglia in situ (Figure 2 and Appendix A). In the adult brain, microglia have a ramified morphology with a small cellular body and multiple long thin processes, which is indicative of their “resting” phenotype. Sholl analysis revealed that p27WT microglia exhibit a highly ramified morphology, while p27KO and p27CK^−^ microglia are less ramified with shorter processes and fewer process tips (Figure 2 and Appendix A), a phenotype also visualized in vitro (Figure 1c). In addition, p27CK^−^ microglia harbor a lower number of branch points, a result not observed in p27KO microglia. Since microglial distribution and density were not affected upon p27 disruption (Appendix A), this reduced morphological complexity is not related to spatial restriction. However, the decrease in morphological complexity is likely to result from cell-intrinsic changes in actomyosin contractility or MT stability as previously reported [30,38,58].

The observed phenotypic characteristics might alter microglial dynamics considering that the cells rely on cytoskeletal rearrangements to carry out their functions [19,41]. Microglial movement is characterized by three different features: (i) cell migration with cell body translocations, (ii) microglial chemotaxis correlated with directed process extension, and (iii) homeostatic activity with continuous extension and retraction of branches (often referred to as microglial branch motility or brain surveillance) [3,41]. These dynamical features are dependent on the microenvironment of microglia as well as cell-intrinsic mechanisms that promote rearrangements of the cytoskeleton [17,41,59]. Moreover, brain surveillance is also regulated by the density of microglia, the velocity of process movement, the extension and retraction frequency, as well as the morphological complexity [17,41].

p27 regulates cell migration in numerous tissues in both physiological and pathological conditions by inducing cytoskeletal remodeling [34,38,39]. Despite this broad role of p27 in cell migration, we observed no significant changes in microglial migration or branch motility upon the constitutive loss of p27 or the expression of its cell cycle dead variant (Figure 3 and Appendix A). The latter results highlight that although morphological complexity influences microglial branch motility, microglia with fewer and shorter branches can exhibit normal brain surveillance [41]. Upon laser-induced brain damage, microglia immediately extended their processes towards the lesion. Yet, the depletion of p27 did not affect the process extension speed (Figure 5). However, the expression of the cell cycle dead variant (p27CK^−^) decreased process extension towards the lesion compared to their p27WT counterparts. This effect might be due to a more stable p27 present in p27CK^−^ microglia, hence likely resulting in a reduced process extension rate, a process highly dependent on MT polymerization.

Constitutive absence of p27 in all brain cells could lead to non-cell-autonomous changes in microglial migration parameters. In particular, the lack of p27 leads to disrupted neuronal migration and axonal transport, an important feature for proper neuronal functioning [19,24,38,39]. On the other hand, microglia express a wide range of receptors, e.g., of the purinergic receptor family P2X or P2Y, that can sense neuronal activity and regulate microglial dynamics [16]. Moreover, Umpierre et al. showed that microglial dynamism is highly affected by bi-directional changes in neuronal activity [11,46]. Therefore, impaired microglial movements might also result from impairments in neuronal activity or migration due to the neuronal absence of p27. To overcome this effect, we generated a conditional knockout mouse (cKO) with a specific depletion of p27 in microglia upon tamoxifen injection. Although we found that the recombination of the p27 gene occurred in mice exposed to an injection of tamoxifen in utero (Appendix A), microglial migration speed and idling time were not affected (Figure 4). Nevertheless, a significant increase in idling speed, i.e., the average instantaneous speed below the idling threshold, and a decrease in active instantaneous speed, i.e., the average instantaneous speed above the idling threshold, were observed in p27cKO microglia compared to p27WT microglia (Figure 4). The percentage of cells contributing to the significant effect of active migration is rather low. However, we cannot exclude that this delayed active speed correlates with disrupted entry and dispersion into the developing CNS, since microglia require this active speed to reach their final position [3,60,61]. The finding that p27 differentially influences the speed of idling and active migration events suggests that further investigation is needed to unravel the involvement of p27 in the specific mechanisms underlying microglial idling and active migration. In breast cancer cells, p27 expression is shown to be inversely linked to the expression of Arpin (Arp2/3), a regulator of idling induction and consequently migration speed, cell turning, and directional persistence [62]. Additionally, the importance of p27 in active cell migration has already been shown in a variety of cell types including neurons, vascular smooth muscle cells, and endothelial cells [21,38,39,63,64]. In particular, in neuronal cells, p27 expression is correlated with alterations in active migration via pathways unrelated to cell cycle progression, i.e., inactivation of RhoA/ROCK activity and modulation of the microtubule and actomyosin cytoskeleton [24,33,38,39,65,66]. It is intriguing to speculate that the significance observed in the active migration and idling speed is of clinical relevance. First, in the embryonic brain, a decreased active migration speed may lead to defects in microglial brain colonization. Microglial cells do acquire this active migration speed to reach their final position in the brain. Alterations in microglial positioning may affect brain development, i.e., neurogenesis, neuronal circuit development, or cell survival [67]. Secondly, in a pathological context, microglial cells require extensive cell body migration to promote both CNS damage and repair [68]. Alternatively, microglial idling is of importance for scanning the environment and hereby contributes to maintaining CNS homeostasis by detecting invading pathogens, dying or dead cells as well as cellular debris [3,50]. Taken together, these data suggest that the conditional loss of p27 alters the active migration of microglia.

Since the constitutive absence of p27 is not affecting microglial migration and surveillance, compensatory mechanisms may be at play. Amongst candidates are p21^cip1^ and p57^kip2^, two other proteins that belong to the Cip/Kip family [69,70]. In contrast to p57, p21 is strongly expressed in microglia [58]. This protein is involved in tumor metastasis and cell motility by relocating to the cytoplasm, inhibiting the Rho pathway, and modulating actin cytoskeleton organization, which makes it a suitable candidate to overtake p27 functions [21,24,71]. In addition, it is interesting to consider the synergetic role of p27 and p21 as research has shown that p27 can indirectly regulate p21 expression and function in mouse embryonic fibroblasts. Yet, this regulation might be cell-type dependent [72]. Protein mass spectrometry or RNA sequencing could determine other possible candidates associated with MTs or actin that might promote or inhibit microglial movement.

### 3.2. Emerging Roles for p27^kip1^ in Microglial Phagocytosis

Microglia are the professional phagocytes of the brain and ensure tissue homeostasis by interacting with infiltrating immune cells, clearing out cellular debris, accumulating metabolic products, and eliminating synapses [5,41,73]. Under pathological conditions, microglia become activated and their processes expand towards the damaged region within minutes. Along with process outgrowth, processes will enlarge and bulbous tips will develop [15]. Directed process motility as a result of such an inflammatory reaction is often followed by further cell activation; cells become less ramified, larger somas appear and phagocytosis occurs to restore homeostasis [17,73]. The deformation of the plasma membrane necessary for phagocytosis arises together with rearrangements of the cytoskeleton in order to engulf particles, synapses, or dead cells. Given the previous reports showing the involvement of p27 in (i) cytoskeletal dynamics by inactivating the RhoA pathway, a pathway also involved in cytoskeletal rearrangements during phagocytosis, and (ii) autophagy, another lysosomal clearance pathway with similar mechanistic and functional properties as phagocytosis, we hypothesized a functional role for p27 in microglial phagocytosis [43,74,75,76,77]. Research in Müller glia and retinal pigment epithelium cells reported that upon the loss of p27, the phagocytic activity was enhanced due to accumulation of phosphorylated myosin light chain II (pMLC-II) [78]. This is consistent with previous research showing higher pMLC-II concentrations in interneurons upon the constitutive absence of p27, which binds to RhoA and prevents its activation by guanine nucleotide-exchange factors (GEFs) [38]. Our study reveals an increased percentage of phagocytic cells in p27KO microglia, while the expression of p27CK^−^ decreased the phagocytic activity (Figure 6). The mechanisms underlying enhanced phagocytosis in the absence of p27 remain elusive; however, in light of previous studies, we speculate that an aberrant activation of the RhoA/ROCK signaling pathway upon the loss of p27 is leading to a higher pMLC-II concentration which, in turn, increases the phagocytic capacity. Consequently, a p27 cell cycle dead variant, which is shown to be slightly more stable than its p27WT counterpart, may lead to lower concentrations of pMLC-II via a stronger inhibition of the RhoA/ROCK signaling pathway [33]. In addition, research in microglial cells proved that blocking this particular pathway decreased phagocytosis, further supporting this hypothesis [79]. Although our speculation might be in contradiction with previous results showing that ablation of RhoA triggers microglial activation, decreases microglial ramification, and produces a neurological phenotype that includes memory deficits, the formation of amyloid-*β* plaques, and the loss of synapses and neurons, hence likely altering the phagocytic capacity of the cells, there is an important difference to consider [80]. Full ablation of RhoA is likely to alter multiple upstream and downstream effectors such as focal adhesion proteins, Arp2/3, or other cytoskeletal proteins such as vinculin or cofilin that might be involved in affecting microglial ramification, motility, and phagocytic activity. Furthermore, we only consider the effect of a cell cycle dead variant or of the loss of p27, which is just one upstream effector of RhoA, hereby excluding the effect of other regulators of this pathway. Yet, it remains interesting to explore the impact of p27 expression levels within this neurological phenotype in microglia caused by the loss of RhoA. In particular, p27 can be phosphorylated by the oncogenic kinase Src, which in turn induces p27 degradation [81]. Soccodato et al. showed that loss of RhoA in microglia triggers Src activation, thereby inducing an amyloid-like pathology [80]. In light of these results, one can speculate that the activation of Src promotes the degradation of p27, which in turn leads to a decrease in the phagocytic capacity of microglial cells, thereby contributing to Alzheimer’s disease. Alternatively, it would be interesting to test whether depletion of p27 is correlated with changes in the expression of phagocytic proteins, such as myosin light chains, considering previous reports that suggest the role of p27 as a transcriptional regulator in several human cancers as well as in Parkinson’s disease [28,82]. Since excessive phagocytosis by microglia could disturb brain homeostasis, future work would be useful to define the underlying mechanism of phagocytosis upon the constitutive absence of p27.

## 4. Conclusions

An extensive examination of various cellular parameters indicated that the depletion or the functional mutation of p27 affects the morphology of microglia, hence hampering dynamic processes in those cells, such as active migration speed, directed branch extension, and phagocytosis. A very interesting aspect is that human and mouse p27 sequences are identical for 92%, including functional motifs such as RhoA binding sites, hereby suggesting similar functions for p27 in mice and humans. [28]. The role of p27 has already been extensively studied in cancers, and its participation in neurodegenerative disorders is strongly suggested. Therefore, considering the importance of p27 in distinct brain cells, it is crucial to know if altered levels of p27 have particular effects on microglial function, since the dysfunction of microglial cells is linked with several neuropathological diseases such as Alzheimer’s disease or Parkinson’s disease [28,72]. Therefore, this research may contribute to new insights into the pathological mechanisms leading to such neurodegenerative disorders.

## 5. Materials and Methods

### 5.1. Contact for Reagents and Resource Sharing

Further information and requests for reagents may be directed to and will be fulfilled by Lead Contact, Bert Brône (bert.brone@uhasselt.be).

### 5.2. Experimental Model and Subject Details

#### Mouse Genetics

All animal experiments were conducted following protocols approved by The Ethical Committee of Université de Liège (Protocol n° 19-2093) or Hasselt University (Protocol n° 202010). Animals were group-housed (unless stated otherwise) in a temperature and humidity-controlled room with ad libitum access to food and water and a 12 h light–dark cycle. For all experiments (in vitro and ex vivo), mice were age matched and combined with littermate controls. Data from male and female mice were pooled as gender-specific effects were not observed.

Generation of the transgenic mice has been previously reported: p27 knockout (p27KO) by [25] and p27 mutant (p27CK^−^) by [34]. CX3CR1^eGFP/eGFP^ [83] mice were obtained from the European Mouse Mutant Archive (EMMA) institute with the approval of Steffen Jung (Weizmann Institute of Science).

CX3CR1^eGFP/+^ p27KO and CX3CR1^eGFP/+^ p27CK^−^ were obtained by first breeding p27KO or p27CK^−^ mice with CX3CR1^eGFP/eGFP^, followed by crossing p27^KO/+^CX3CR1^eGFP/eGFP^ or p27^CK/+^CX3CR1^eGFP/eGFP^ mice with heterozygous p27 obtained from the KO or CK^−^ mouse line. p27 floxed (p27^Fl^) mice were generated as described previously [84]. The CX3CR1^CreER(T2)^ mice were obtained from the EMMA institute [85]. p27^Fl/+^CX3CR1^CreEr(T2)/+^ were obtained by breeding p27^Fl/Fl^ mice with CX3CR1^CreEr(T2)/CreEr(T2)^. p27 conditional KO (p27^Fl/Fl^CX3CR1^CreEr(T2)/+)^) embryos that were used in this study were obtained by time mating female p27^Fl/+^CX3CR1^CreEr(T2)/+^ with male p27^Fl/Fl^ mice overnight. Females with a vaginal plug in the morning were considered pregnant and henceforth designated with embryonic day 0.5 (E0.5). Pregnant females were housed individually. Recombination was induced by injecting pregnant females subcutaneously at E12.5 and E14.5 with 2 mg/35 g body weight of tamoxifen (T5648, Sigma-Aldrich, Overijse, Belgium, 20 mg/mL in corn oil). Pregnant mice were sacrificed by cervical dislocation at E15.5 and embryos were surgically dissected.

### 5.3. Method Details

#### 5.3.1. Primary Microglia Isolation and Culture

Primary microglia were isolated from p27KO and p27CK^−^ mouse brains. Brains were processed separately to prepare the cells and to generate biological replicates. Briefly, brains were dissected from postnatal day 3 (P3) pups, and meninges were carefully removed. Brains were mechanically dissociated using pipettes of decreasing diameter and the homogenate was filtered using a 70 µm cell strainer (Thermo Fisher Scientific, Waltham, MA, USA). After centrifugation for 5 min at 300× *g* at 4 °C, cell pellets were resuspended in Dulbecco’s Modified Eagle Medium (DMEM, Sigma-Aldrich) supplemented with 10% fetal bovine serum (FBS, Gibco^TM^, Waltham, MA, USA), 10% horse serum (HS, Gibco^TM^), and 1% Penicillin/Streptomycin (P/S, Gibco^TM^, hereafter referred to as DMEM 10:10:1) preheated at 37 °C and cultured for 7–10 days in a humidified atmosphere of 95% air and 5% CO_2_. After 10 days, the culture medium was replaced with fresh DMEM 10:10:1 medium supplemented with 30% colony stimulating factor-conditioned medium obtained from L929 cells for 3–5 days. L929 cells were cultured in DMEM supplemented with 10% FCS (Gibco^TM^), 1% P/S, 1% L-Glutamine (Thermo Fisher Scientific), and 1% non-essential amino acids (Sigma-Aldrich) for 14 days before collecting and filtering the colony-stimulating factor-conditioned culture medium. Shake-off was performed for 3 h at 230 rpm, at 37 °C. Microglial cells were filtered through a 70 µm cell strainer and centrifuged for 10 min at 300× *g* after which the cell suspension was resuspended in DMEM 10:10:1 and seeded for experiments.

#### 5.3.2. Immunofluorescence

Primary microglia were seeded on poly-D-lysine (PDL, #PD6407, Sigma-Aldrich)-coated coverslips for 24 h (2 × 10^4^ cells/cover glass) before staining. Cells were fixed with 4% paraformaldehyde (PFA, Sigma-Aldrich) for 15 min, at room temperature (RT), washed three times in PBS containing 0.3% Tween-20 (Sigma-Aldrich) and 0.2% Triton X-100 (Sigma-Aldrich), blocked and permeabilized in 10% donkey serum, 0.3% Tween-20 and 0.2% Triton X-100 in PBS for 1 h, at RT, and incubated with primary antibodies overnight, at 4 °C. The cells were then washed 3 times in PBS containing 0.3% Tween-20 and 0.2% Triton X-100 and incubated with Alexa-labeled secondary antibodies (Thermo Fisher Scientific) for 1 h at RT. Cells were then washed in PBS, incubated in DAPI (Sigma-Aldrich), and mounted on glass slides using Dako Fluorescence mounting medium (Dako, Machelen, Belgium). All images were acquired using a 60× oil immersion objective (NA 1.4) on a Nikon A1R confocal microscope. Staining controls for the secondary antibodies were performed by omitting the primary antibodies.

To determine p27 expression in microglia, cells were incubated with rabbit anti-p27 (1:300, 25614-1-AP, Proteintech, Manchester, UK) and goat anti-Iba1 (1:300, ab5076, Abcam, Cambridge, UK) primary antibodies. A Ki-67 antibody (1:250, #550609, BD Biosciences, Erembodegem, Belgium) was used to identify active proliferating microglial cells. The antigen is expressed during all active phases (the G1, S, G2, and M phases) of the cell cycle and cannot be detected in resting cells (G0 phase) [86].

#### 5.3.3. Western Blotting

Freshly isolated primary microglia were lysed on ice in cold RIPA lysis buffer containing: 50 mM Tris, 150 mM NaCl, 1 mM EDTA, 1% IGEPAL CA-630 and 0.50% Na^+^ deoxycholate supplemented with protease inhibitor (Roche, Basel, Switzerland). Proteins were collected from the supernatant upon centrifugation and concentrations were determined using the bicinchoninic acid protein (BCA) assay kit (#23225, Thermo Fisher Scientific). Equal protein amounts (diluted in Laemmli buffer with 5% β-mercaptoethanol and heated for 4 min, at 95 °C) were used for SDS-PAGE (12% SDS-PAGE gel). Gels were transferred to a polyvinylidene fluoride (PVDF) membrane and blocked for 2 h with blocking buffer (Tris-buffered saline-0.1% Tween-20 containing 5% skimmed milk (Carl Roth, Karlsruhe, Germany)). Rabbit anti-p27 antibody (1:500, 25614-1-AP, Proteintech) was subsequently incubated overnight, at 4 °C, followed by washing steps and incubation with a horseradish peroxidase-conjugated secondary antibody (#G-21040, Invitrogen, Merelbeke, Belgium) for 1 h. All antibodies were diluted in blocking buffer, and incubations were at RT unless stated otherwise. Proteins were visualized using the enhanced chemiluminescence system (#32106, Pierce^TM^ ECL Western Blotting substrate, Thermo Fisher Scientific) and ImageQuant LAS 4000. Homogenous loading was checked using α-tubulin (#T9026, Sigma-Aldrich) staining.

#### 5.3.4. Scratch Migration Assay

Scratch migration assays were performed according to the instructions of the IncuCyte^®^ manufacturer. Briefly, cells were seeded into a 96-well ImageLock plate (Essen BIOSCIENCE, Newark, UK) at a density of 1.6 × 10^5^ cells/well and placed in an incubator overnight, at 37 °C, in a humidified atmosphere with 5% CO_2_. The cell monolayer was scratched using the 96-pin IncuCyte WoundMaker^TM^ (Essen BIOSCIENCE), washed three times in PBS, and incubated in fresh DMEM 10:10:1 preheated at 37 °C. Images were taken every 1 h over a total duration of 24 h using the IncuCyte live-cell analysis system and a 10× IncuCyte^®^ ZOOM objective (Cat. No. 4464). Relative scratch densities were calculated using the Incucyte^®^ Base Analysis Software.

#### 5.3.5. Quantification of p27^kip1^ Expression in Microglia after Pro- and Anti-Inflammatory Stimulation

Primary microglia were seeded at a concentration of 10^5^ cells/well in a 96-well plate pre-coated with PDL (20 µg/mL, Sigma-Aldrich) and incubated for 24 h, at 37 °C, in a humidified incubator with 5% CO_2_. Afterward, culture medium was replaced with 100 µL DMEM 10:10:1 (Control) or 100 µL DMEM 10:10:1 containing a pro- or anti-inflammatory stimulus, all from Peprotech (Rocky Hill, NJ, USA) except if stated otherwise (IL-4 (33 ng/mL, #214-14), IL-13 (33 ng/mL, #210-13), TNFα (25 ng/mL, #315-01A), IFNγ (100 ng/mL), LPS (1 mg/mL, #437625, Merck Millipore, Overijse, Belgium), or CSF (30%, obtained from L929 cells) preheated at 37 °C. After 24 h, the mRNA of microglia was extracted via the Qiazol method using the RNeasy mini kit (#79306, #74106, Thermo Fisher Scientific) and stored immediately at −80 °C. RNA was converted into cDNA using qScript cDNA SuperMix (#95048, Quantabio, Beverly, MA, USA). Quantitative RT-PCR (qRT-PCR) was performed using the SYBR Green master mix (#4385612, Applied Biosystems, Waltham, MA, USA). qPCR measurements were expressed as Ct values (determined by the second derivative method). For each condition, these Ct values were used to determine the relative expression of the gene of interest (normalized to the control) compared to housekeeping genes using the 2^−ΔΔCt^ method as each gene was amplified with similar efficiency. To verify p27 expression, the following primer sequences were used (IDT, Leuven, Belgium): 5′-GAATCATAAGCCCCTGGAGGGC-3′ (forward) and 5′-AATGCCGGTCCTCAGAGTTTGC-3′ (reverse). 18S ribosomal RNA was used as a housekeeping gene with the following primer sequences: 5′-ACGGACCAGAGCGAAAGCAT-3′ (forward) and 5′-TGTCAATCCTGTCCGTGTCC-3′ (reverse).

#### 5.3.6. Nitric Oxide Assay

Primary microglia were seeded at a concentration of 10^5^ cells/well in a 96-well plate pre-coated with PDL (20 µg/mL, Sigma-Aldrich) and incubated for 24 h, at 37 °C, in a humidified incubator with 5% CO_2_. Afterward, the culture medium was replaced with 100 µL DMEM 10:10:1 (Control) or 100 µL DMEM 10:10:1 containing 250 ng/mL IFNγ (Peprotech) preheated at 37 °C. After 24 h, supernatants were collected and the nitrite concentration, a breakdown product of nitric oxide (NO), was measured using the Griess reagent kit (#G2930, Promega, Leiden, The Netherlands) following the manufacturer’s instructions.

#### 5.3.7. Phagocytosis Assay

The phagocytic capacity of cultured microglia was evaluated using DiI-labeled synaptosomes. Synaptosome purification was performed using the Syn-Per^TM^ synaptic protein extraction reagent (#87793, Thermo Fisher Scientific). For this, neuronal tissue from P21 C57BL/6 wild type (WT) mice was homogenized in 4 mL Syn-Per reagent and centrifuged for 10 min at 1200× *g*, at 4 °C. Homogenate flow-through was collected and synaptosomes were spun down for 20 min at 15,000× *g* and 4 °C. Synaptosomes were resuspended in 500 µL Syn-Per reagent, labeled with 1,1′-Dioctadecyl-3,3,3′,3′-tetramethylindocarbocyanine perchlorate (DiI), and stored at −80 °C until further use. Primary microglia were seeded at a concentration of 10^5^ cells/well in a 96-well plate pre-coated with PDL (20 µg/mL, #468495, Sigma-Aldrich) and incubated for 24 h at 37 °C in a humidified incubator with 5% CO_2_. DiI-labeled synaptosomes (3 µg/well) were added to the culture media and the cells were incubated for an additional 10 min up to 2 h. Finally, cells were washed and phagocytic activities were determined by measuring DiI-fluorescence using a BD FACSCanto II Cytometer. The percentage of phagocytic cells was calculated by means of cell fluorescence following synaptosome uptake. In addition, fluorescence intensity differences were used to determine the amount of synaptosome uptake in phagocyting microglia (shown as mean fluorescent intensity, MFI). Data were analyzed using FlowJo^TM^ Software (BD Biosciences, https://www.flowjo.com, accessed on 10 February 2020).

#### 5.3.8. Sholl and Microglial Density Analysis in Fixed Brain Slices

CX3CR1^eGFP/+^ p27KO, CX3CR1^eGFP/+^ p27CK^−^ and CX3CR1^eGFP/+^ p27WT mice were terminally anesthetized at P21 by intraperitoneal injection of 2.5 mg/g (body weight) of Dolethal (#570681.8, Vetoquinol, Aartselaar, Belgium). Mice were transcardially perfused with cold PBS containing heparin (20 I.U., Heparin LEO 5.000 I.U./mL, Lier, Belgium)) followed by 4% cold PFA. Brains were dissected and further post-fixed in 4% PFA overnight, at 4 °C, washed in PBS, and kept in PBS-azide (0.01%) until slicing. Free-floating sections (100 µm) were obtained using a Microm HM650V Vibratome (Prosan, Monheim, Germany) and stained with DAPI (Sigma-Aldrich) for 15 min. Thereafter, the sections were mounted between microscope slides (Thermo Fisher Scientific) and coverslips in fluorescent mounting medium (Immuno-Mount, Thermo Fisher Scientific). Cortical sections were imaged using a confocal microscope (ZEISS LSM880) and eGFP-expressing microglia were visualized using an Argon 488 nm laser. Microglial cells were captured within a z-stack of 20 µm (1 µm step) using a 63× oil objective (NA 1.4) at 512 × 512 pixels. For surface density analysis, cortical brain sections were imaged using an automated slide scanner (Zeiss AxioScan.Z1) fit with a 20× objective.

Microglial morphology was assessed by Sholl analysis, as originally described by Sholl and according to previous research [44,45,87,88]. Briefly, cell reconstructions were performed using 3D automatic cell tracing in Vaa3D software (http://www.vaa3D.org, accessed on 1 February 2021). The software runs the APP2 (All-path-pruning 2.0) algorithm to generate ramified cells in 3D [89]. Custom codes in MATLAB (Mathworks Inc, https://www.mathworks.com accessed on 1 February 2021, available at https://github.com/AttwellLab/Microglia, accessed on 1 February 2021) were used to derive morphological parameters. The reader is referred to the above-mentioned paper for more details. Microglial surface density was quantified in the primary and secondary motor cortex layer 1, 2/3, and 5 by calculating the number of cells per area unit (mm^2^). The volumetric density of microglia in the cortex was studied by manually counting cells using Fiji software [90]. Analyses were carried out with the experimenter blinded to the genotype.

#### 5.3.9. Microglial Dynamics in Acute Adolescent Brain Slices Using Two-Photon Imaging

The study of microglial dynamics in acute adolescent brain slices using two-photon imaging was performed according to previous research [88]. P21 brains were dissected and incubated into ice-cold slicing solution containing (in mM): 120 N-methyl-D-glucamine, 2.5 KCL, 25 NaHCO_3_, 1 CaCl_2_, 7 MgCl_2_, 1.2 NaH_2_PO_4_, 20 D-glucose, 2.4 Na^+^ pyruvate, 1.3 Na^+^-L-ascorbate, pH 7.3–7.4, ~300 mOsm, which was constantly perfused with 95% O_2_ and 5% CO_2_. Brains were coronally sliced (300 µm thick) using a vibratome (LEICA VT1200S) and allowed to recover for 1 h, at 36 °C, in oxygenated artificial CSF (aCSF) containing (in mM): 126 NaCl, 2.5 KCl, 26 NaHCO_3_, 2 CaCl_2_, 2 MgCl_2_, 1.25 NaH_2_PO_4_, 10 D-glucose, pH 7.3–7.4, ~300 mOsm. Experiments were performed under continuous perfusion of oxygenated (95% O_2_/5% CO_2_) aCSF at RT to preserve slice health. A Zeiss LSM880 confocal microscope (with a 40× EC plan-Neofluar objective, NA 1.4) provided with a Mai Tai DeepSee Ti:Sapphire pulsed laser (Spectra-Physics, Utrecht, The Netherlands) tuned at 920 nm with a pixel dwell time of 1.54 µsec was used to image eGFP-positive microglia in the cortex. Stacks were recorded starting from a minimum depth of 50 µm beneath the surface of the slice to avoid cells activated by slicing [91,92]. For branch motility analysis, a z-stack spanning 14 µm with serial optical sections at 512 × 512 pixels using a 16-bit line averaging every 1 µm was acquired as a time series. For imaging, the laser power was adjusted to ~13 mW. Ablation of a small tissue volume was induced by illuminating a 5 µm radius area for 1 min with the laser power increased 8-fold (110 mW, 65 µsec pixel dwell). To image directed process motility in response to the laser injury, a z-stack spanning 14 µm with a 1 µm depth interval was acquired every 60 sec for a total duration of 15 min. Images were typically 512 × 512 pixels and covered a square field of view of 212 µm × 212 µm.

All images were processed using Fiji software as described before [15,87,90]. For every image, each slice of every stack was filtered with a median filter (value 1) after subtraction of background noise. The stacks were then registered for drifting by applying the StackReg plugin with rigid body transformation [93]. For quantification and analysis of microglial surveillance, maximum intensity projections were performed and individual cells were selected by manually drawing a region of interest and erasing data outside this region. These individual cells were then manually binarized by applying the Huang threshold which is based on the intensity and morphology of the cells. Threshold values were set ensuring the presence of all microglial processes in all the different frames. To quantify surveillance, for each movie, consecutive and binarized images were pairwise subtracted to generate a new movie consisting of pixels containing process extension and retraction. The scanned area was then calculated as the sum of these pixels over a time span of 10 min. For analysis and quantification of microglial-directed process motility, the resulting movies were processed as described above and analyzed using the MTrackJ plugin by tracking groups of processes from every individual cell reacting to the lesion [94]. Average migration speed (µm/min) was calculated as the total length of the traveled path divided by the duration of the track and averaged per cell. Instantaneous speeds (µm/min), i.e., the speed between consecutive time points, were calculated per cell and plotted as a function of time.

#### 5.3.10. Microglial Migration in Acute Embryonic Brain Slices Using Two-Photon Imaging

Embryonic brains from E15.5 mice were isolated in ice-cold PBS-glucose (pH 7.4; 25 mM glucose) and embedded in 4% low melting agarose (Thermo Fisher Scientific). Prior to slicing, genotyping was performed to select p27 conditional KOs (cKO) as well as their littermate wild types (WT). Brains were coronally sliced at a thickness of 300 µm using a vibratome (LEICA VT1200S). Brain slices were transferred on MilliCell organotypic inserts (Merck Millipore) in a 24-well plate designed for confocal microscopy (IBIDI, Gräfelfing, Germany) and equilibrated in semi-hydrous conditions, at 37 °C and 5% CO_2_. Migration media consisted of Neurobasal medium supplemented with 2.5 mM isolectin GS-IB4-Alexa 488 (Life Technologies, #I21411, Carlsbad, CA, USA) to visualize microglia, 2 mM L-glutamine (#25030081), 1× B27 supplement (#17504-044), 1× N2 supplement (#17502-048), and 0.5% P/S (All from Thermo Fisher Scientific).

Image acquisition started 1 h after tissue resting at 37 °C and 5% CO_2_ and was completed within 8 h after brain dissection. During measurements, the environment chamber of the microscope was heated by constant air provision, at 37 °C, and 5% CO_2_ humidified air was applied directly onto the slice. A confocal microscope (ZEISS LSM880) provided with a Mai Tai DeepSee Ti:Sapphire pulsed laser (Spectra-Physics) tuned at 780 nm and a 20x EC plan-Neofluar objective (NA of 0.5 and 2 mm working distance) was used to image isolectin GS-IB4-Alexa 488-labeled microglia in the embryonic cortex. A z-stack spanning 72 µm (8 µm interval) with serial optical sections at 1024 × 1024 pixels was acquired every 10 min for a total duration of 6 h. Stacks were recorded starting from a minimum depth of 50 µm beneath the surface of the slice to avoid cells activated by slicing [91,92]. Image processing and cell migration tracking were performed using Fiji software, as described by our lab [50]. Time series were first corrected for 3D drift using the 3D drift correction plugin and microglial migration was manually tracked in 3D using MTrackJ [94]. Only cells remaining in the field of view for at least 100 min were included in the analysis. Average migration speed (µm/h) was calculated as the total length of the travelled path divided by the duration of the track. Relative idling time, defined as the percentage of time the cell spent on pausing, was calculated using a custom-made macro developed by Gorelik and Gautreau [49]. The threshold for idling was calculated from the mean squared displacement and set at 15 µm per 10 min. Average instantaneous speeds of the events below (Average V_inst. idling_) and above the idling threshold of 1.5 µm/min (Average V_inst. act_) were calculated as the distance travelled during consecutive frames, divided by the time interval (10 min).

To verify recombination of the p27 gene after tamoxifen administration, a RT-PCR on brain slices from each embryo was performed using the following primer sequences (IDT, Leuven, Belgium): 5′-CTAGGTTTCGCGGGCAAAGA-3′ (forward) and 5′-CTCCCATCCAATTCGACAAC-3′ (reverse).

### 5.4. Quantification and Statistical Analysis

Statistical analyses and graphs were generated using Prism 9 (GraphPad Software, https://www.graphpad.com accessed on 1 January 2018). Data distributions were assessed for normality (Shapiro wilk). In case the assumptions for normality were met for all groups, a Student’s *t*-test was performed. When data distribution of at least one group was non-Gaussian, non-parametric tests such as the (multiple) Mann–Whitney *U* test were used. In Figure 4f, Fisher’s exact test was used to assess non-random associations. Graphs represent median or mean ± SEM dependent on the distribution of the data. *p*-values smaller than 0.05 were considered significant with * *p* < 0.05, ** *p* < 0.01, *** *p* < 0.001 and **** *p* < 0.0001. The reader is referred to the figure legends for details about sample sizes used for statistical analyses.

## Figures and Tables

**Figure 1 ijms-23-10432-f001:**
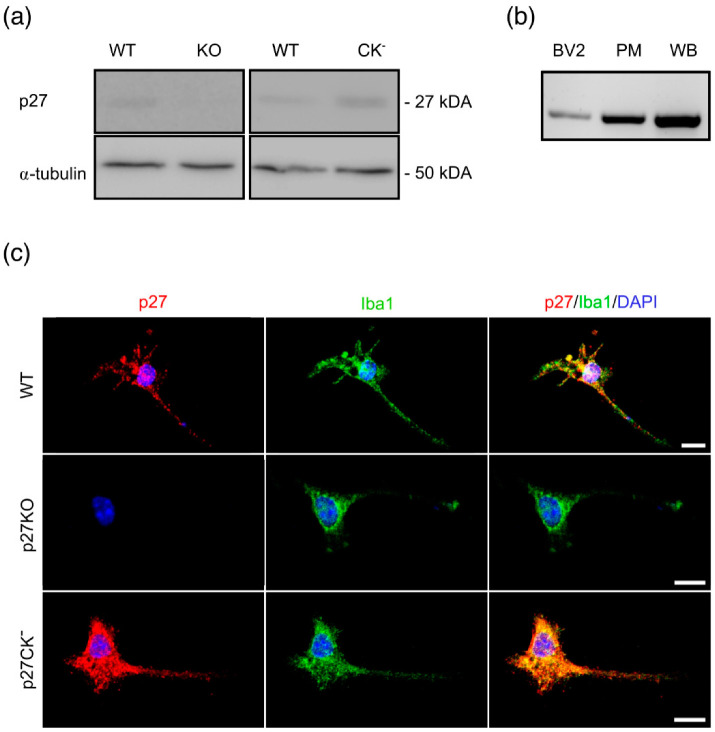
**p27 is expressed in microglial cells.** (**a**) Representative Western blots showing the expression of p27 in primary wild type (WT) and p27CK^−^ microglial cells. α-tubulin was used as a loading control. (**b**) RT-PCR of p27 expression in a murine microglial cell line (BV2 cells), in primary microglial cells (PM), and in a whole brain (WB) extract. (**c**) Immunofluorescent staining of primary p27WT, p27KO, and p27CK^−^ microglia. Cells were stained with an anti-p27 (red) and an anti-Iba1 (green) antibody and their nuclei were counterstained with DAPI (blue). Scale bar = 10 µm. See also Appendix A.

**Figure 2 ijms-23-10432-f002:**
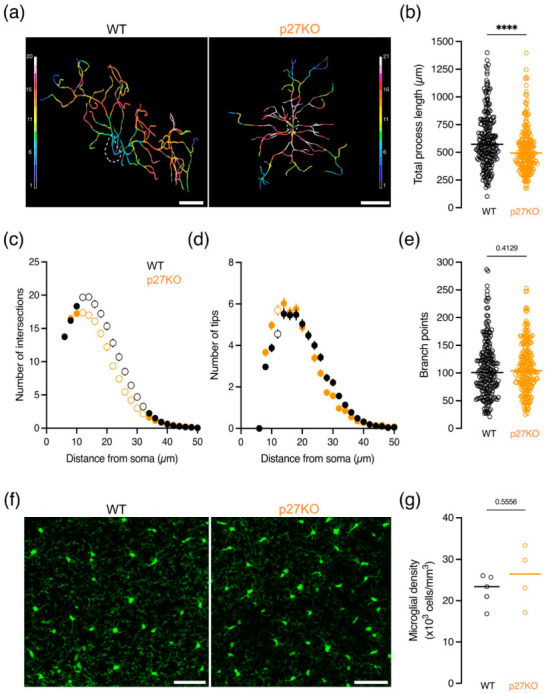
**p27 knockout microglia exhibit less ramified processes.** (**a**–**e**) Morphology analysis of microglia from perfusion-fixed P21 CX3CR1^eGFP/+^ p27WT (230 cells, 5 mice) and CX3CR1^eGFP/+^ p27KO (223 cells, 4 mice) mice showing: (**a**) Representative 3D skeletonized microglia from postnatal day 21 p27WT and p27KO microglia. Color coding represents the number of intersections (soma centered). Scale bar = 10 µm; (**b**) Total process length; (**c**) Number of process intersections at a specific distance from the soma; (**d**) Number of tips (terminal points); and (**e**) Branch points. (**f**) Representative images of perfusion-fixed P21 CX3CR1^eGFP/+^ p27WT and CX3CR1^eGFP/+^ p27KO cortical microglia in brain slices. Scale bar = 50 µm. (**g**) Average microglial density in the cortex of 5 p27WT and 4 p27KO brains. **** *p* < 0.0001. Mann–Whitney *U* test. Graphs (**b**,**e**) represent individual microglia. Horizontal bars represent the median. Empty data points in (**c**,**d**) represent significant differences (*p* < 0.0001). See also Appendix A.

**Figure 3 ijms-23-10432-f003:**
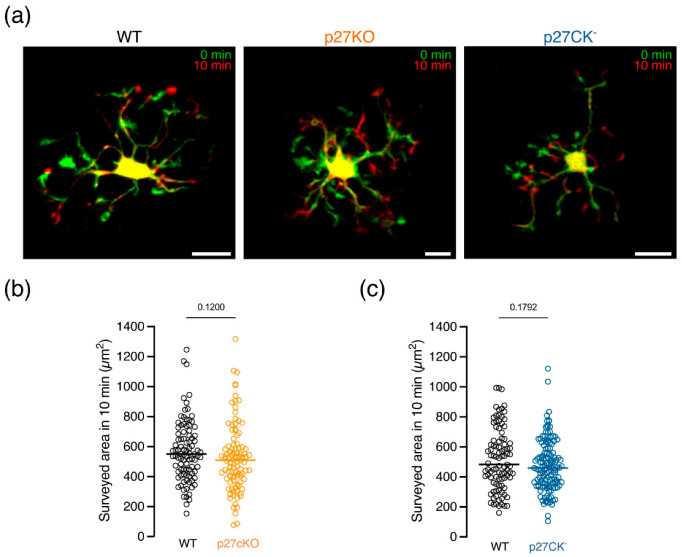
**Microglial surveillance of the cortex does not involve p27.** (**a**) Representative images showing microglial surveillance for CX3CR1^eGFP/+^ p27WT, CX3CR1^eGFP/+^ p27KO and CX3CR1^eGFP/+^ p27CK^−^ microglia. Images taken 10 min apart are overlaid to show microglial movement. (**b**,**c**) The surveyed area is not significantly different for p27KO (**b**) and p27CK^−^ (**c**) microglia compared to p27WT cells. Data points represent individual microglia. Sample size (**b**) for p27WT: 93/8/3 (cells/brains/mothers); p27KO: 95/6/3. Sample size (**c**) for p27WT: 95/6/3; p27CK^−^: 135/8/3. Horizontal bars represent the median. Scale bar = 10 µm. Mann–Whitney *U* test.

**Figure 4 ijms-23-10432-f004:**
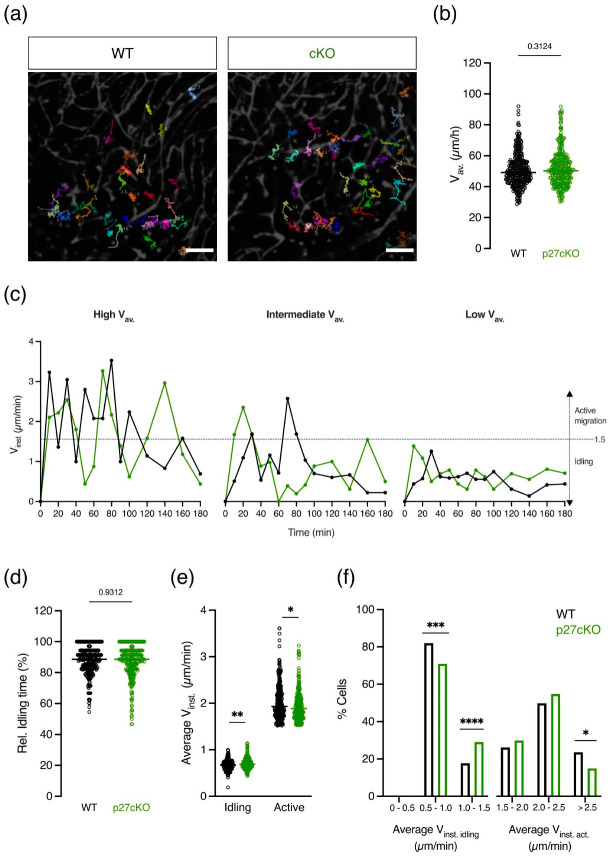
**Conditional loss of p27 affects the instantaneous speed of microglia during phases of active migration.** Microglial movement was recorded in acute brain slices for 6 h using two-photon time-lapse imaging and cell somas were manually tracked. (**a**) Representative microglial migration tracks from p27WT and p27 conditional knockout (cKO) cells. Different colors indicate independent microglia. Scale bar = 100 µm. (**b**) The average migration speed (V_av._) did not change after the conditional loss of p27. Sample size for p27WT: 322/3/3 (cells/brains/mothers); p27cKO: 307/6/3. Mann–Whitney *U* test. (**c**) Representative instantaneous velocity (V_inst._) as a function of time of a cell migrating at high (left panel), intermediate (middle panel), and low (right panel) speed for both p27WT (black) and p27cKO (green). (**d**) Relative idling time is not changed upon conditional loss of p27. Mann–Whitney *U* Test. (**e**) The average instantaneous speeds of both the idling (Average V_inst. idling_) and active migration events (Average V_inst.act._) were significantly changed upon conditional loss of p27. Mann–Whitney *U* test. (**f**) The percentage of cells contributing to the significant differences in both Average V_inst. idling_ and Average V_inst.act_ plotted in (**e**). Fisher’s exact test. Sample size (**b**,**d**) for p27WT: 322/3/3 (cells/brains/mothers); p27cKO: 308/6/3. Sample size ((**e**), idling) for p27WT: 322/3/3; p27cKO: 308/6/3. Sample size ((**e**), active) for p27WT: 271/3/3; p27cKO: 261/6/3. Data points represent individual microglia. Horizontal bars represent the median. * *p* < 0.05, ** *p* < 0.01, *** *p* < 0.001, **** *p* < 0.0001. See also Appendix A.

**Figure 5 ijms-23-10432-f005:**
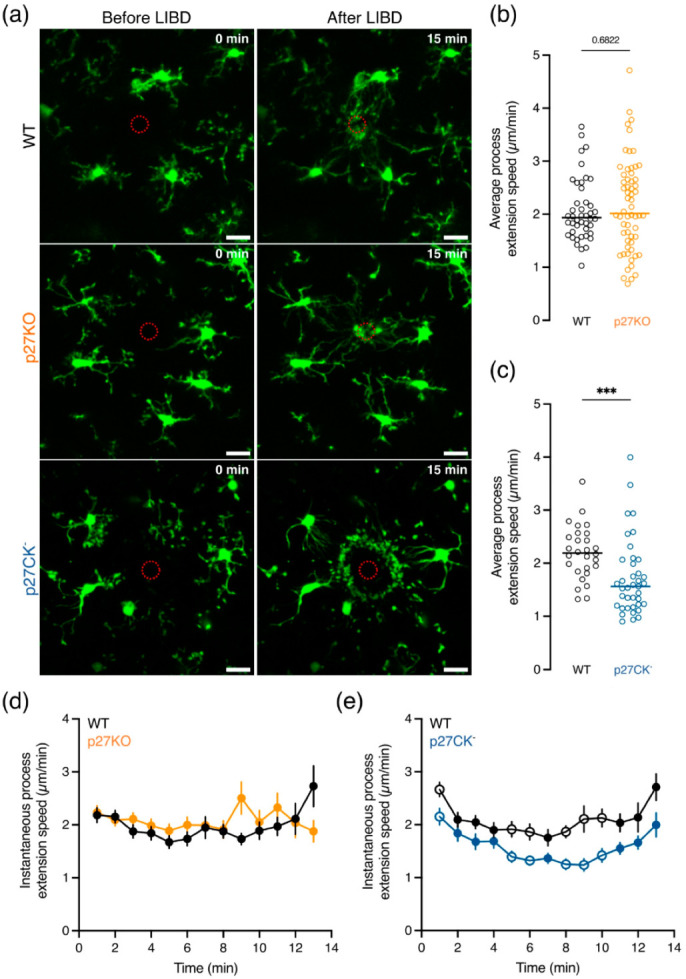
**Process extension speed is decreased in microglia expressing a cell cycle dead variant of p27.** (**a**) Representative images showing cortical microglia ex vivo rapidly responding towards laser-induced brain damage (LIBD). Scale bar = 20 µm. (**b**,**c**) The average process extension speed did not differ between p27WT and p27KO (**b**) microglia while p27CK^−^ (**c**) microglia displayed a slower process extension speed compared to p27WT cells. (**d**,**e**) The instantaneous velocity as a function of time shows no speed differences between p27KO (**d**) and p27WT microglia but a decreased speed upon a cell cycle dead variant of p27 (p27CK^−^) (**e**). Sample size (**b**) for p27WT 44/6/3 (cells/brains/mothers); p27KO 60/5/3. Sample size (**c**) for p27WT 28/3/3; p27CK^−^ 39/3/3. Data points in b and c represent individual microglia and the horizontal bars represent the median. *** *p* < 0.001. Empty data points (in (**d**,**e**)) represent significant differences (*p* < 0.001), and values are reported as mean ± SEM. Mann–Whitney *U* test.

**Figure 6 ijms-23-10432-f006:**
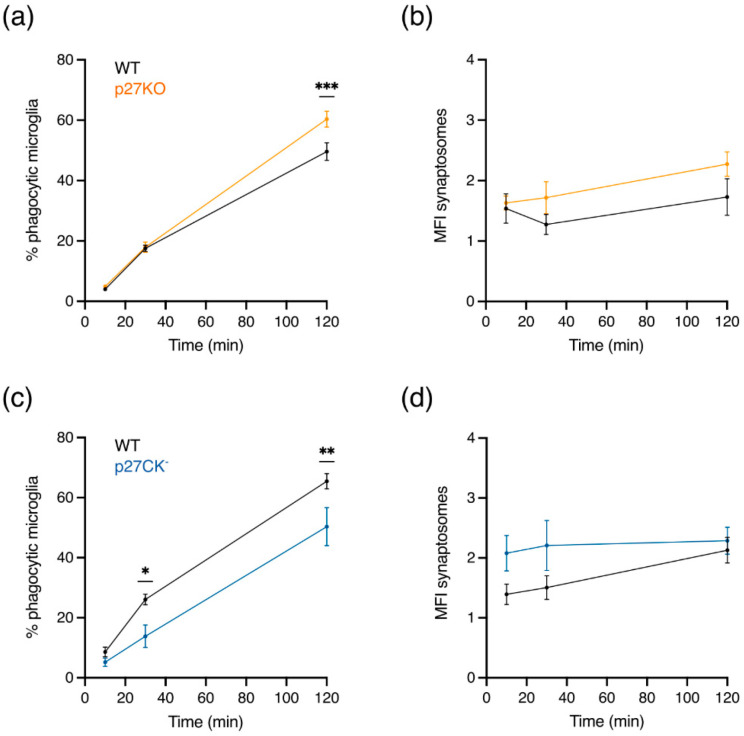
**Primary microglia show a change in phagocytic capacity upon modulation of p27 expression.** (**a**,**b**) The percentage of phagocytic microglia after 2 h incubation with DiI-labeled synaptosomes was significantly increased upon loss of p27 (p27KO) (**a**) but decreased after 30 min and 2 h incubation in the cell cycle dead variant of p27 (p27CK^−^) (**b**). (**c**,**d**) The uptake (median fluorescence intensity, MFI) of DiI-labeled synaptosomes in phagocytic microglia did not significantly change upon loss of p27 (**c**) or a cell cycle dead variant (**d**) of p27. Sample size (**a**,**b**) for p27WT: 9/5 (brains/mothers); p27KO: 9/5. Sample size (**c**,**d**) for p27WT: 9/6; p27CK^−^: 5/6. Values are reported as mean ± SEM. * *p* < 0.05, ** *p* < 0.01, *** *p* < 0.001. Multiple Mann–Whitney *U* test.

## Data Availability

Not applicable.

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
