# Peer review of "p27kip1 Modulates the Morphology and Phagocytic Activity of Microglia"

_ijms, 2022, doi:10.3390/ijms231810432_

Round 1
Reviewer 2 Report
Very well written and good presentation, however, author mentioned RT-PCR in Fig 1 but did not show any. And please use color code to indicate intersections in 3D sholl analysis.
Round 2
Reviewer 1 Report
The authors have addressed most of my comments. The text is substantially improved, and references have been added throughout the text. However, I have some minor concerns that are mentioned below in the attached point-by-point response letter. My comments are shown in orange. Importantly, I would like to know if the compensatory mechanism could be investigated with the existing material. I consider the manuscript ready for publication after performing very minor edits.

Author Response
Dear Prof. Dr. Battino, Dear Mr. Aiden Shang, Dear Editors, Dear Reviewer 1,
We sincerely thank you for the swift follow up of our resubmission and are very grateful to the reviewers for their kind words on the manuscript.
Specifically concerning the second review report of reviewer 1:
All changes to the manuscript are indicated using track changes.
We have carefully re-read the manuscript for additional typical errors and a spell check has been performed to improve the language in the manuscript.
Regarding the inquiry of Reviewer 1 concerning “the evaluation of p21 expression on remaining material (if any)”, we could not accommodate as unfortunately there is not enough material remaining to allow rigorous analysis on the mRNA or protein level. Reviewer 1 confirmed in his/her response to the first review round, that additional mice breeding’s, cell isolation and ethical permission in order to obtain enough material for the experiment would be not in balance with the added value, therefore, the authors decided not to carry on those. We hope the improved version of our manuscript is suitable for publication in IJMS.
Yours sincerely,
Bert Brône on behalf of all co-authors.